# OpenReview forum: "Aligning Deep Implicit Preferences by Learning to Reason Defensively"
_ICLR.cc/2026/Conference — ICLR 2026 Poster_

### Official Review · Reviewer_9KAU · 2025-10-28

**Soundness:** 2
**Presentation:** 2
**Contribution:** 3
**Rating:** 4
**Confidence:** 3

**Summary:**

This paper introduces **Critique-Driven Reasoning Alignment (CDRA)**, a framework designed to bridge the gap between users’ implicit preferences and their explicit, surface-level feedback in personalized alignment. The authors first construct a synthetic dataset, DeepPref, which captures the reasoning process behind inferring users’ latent intents and corresponding rewards. Using this dataset, they train the **Personalized Generative Process Reward Model (Pers-GenPRM)**, which models personalized reward inference as a reasoning task. Finally, they propose **Critique-Driven Policy Alignment (CDPA)**, which leverages the learned rewards to compute token-level advantages and mitigate the zero-gradient issue. Experimental results demonstrate that CDRA consistently outperforms all baselines on both the DeepPref and PrefEval benchmarks.

**Strengths:**

- The motivation of bridging the gap between users’ implicit preferences and surface-level ones is clear and reasonable, effectively framing personalized alignment as a reasoning process.
- The idea of using the reward model to provide token-level advantages is solid, as it offers denser feedback signals when optimizing the policy.

**Weaknesses:**

- The experiments are conducted on only one model; the results would be more convincing if additional models were included.
- The paper lacks a comprehensive human study; directly assessing users’ satisfaction with the responses would strengthen the evaluation.

**Minor issues:**

- It is somewhat confusing and potentially misleading to place the *Evaluation Protocol* paragraph before the training section. I suggest moving it afterward for better readability.
- Some notations are undefined, including $N$ in Equation 1 and $r_{i,t}$ in Equation 5.

**Questions:**

- How does the model identify the $T_i$ reasoning steps within each response?
- Regarding the proposed metric, *Deep Preference Understanding*, does the prompt for the LLM evaluator of step-wise rewards and critiques include similar criteria? If so, could this lead to reward hacking?
- The dataset appears to contain only surface-level preferences, not implicit ones. How can we be sure that the LLM evaluator truly captures users’ latent intents rather than hallucinating them?
- Since the reward model (Pers-GenPRM) is trained in a supervised manner, is it possible to directly use the LLM evaluator without fine-tuning a separate reward model?

---

> ### Author Response · Authors · 2025-11-20
> **Responses to the Reviewer 9KAU [1/4]**
>
> We deeply appreciate your high praise for the core motivation of this work—reframing personalized alignment as a process of deep reasoning. Your recognition serves as a significant encouragement to us.
>
> Below is our detailed response to your questions:
>
> **W1: Limited Model Diversity**
>
> **R1:** We appreciate you pointing this out. To demonstrate that our motivation translates into robust and generalizable performance gains, we have conducted extensive supplementary experiments:
>
> 1.  **Cross-Model Generalization:** To demonstrate that CDRA is not dependent on a specific model architecture, we extended our experiments to include **Llama-3-8B-Instruct** and additional variants within the **Qwen** series. As shown in the updated **Table 1**, CDRA achieves consistent and significant performance improvements across all tested architectures.
>
> 2.  **Comparison with Alternative Paradigms:** We conducted a direct comparison between CDRA and mainstream reward paradigms, including Process-level Scalar Rewards (PRM), Outcome-level Generative Rewards (GRM), and Rubric-based methods. The experimental results (**Table 2**) indicate that our method significantly outperforms these alternatives in deep preference inference tasks.
>
> In summary, these supplementary experiments strongly demonstrate that our proposed framework of "viewing alignment as reasoning" is not only theoretically sound but also practically efficient and robust. Once again, we thank you for your recognition, which has greatly encouraged us.
>
> ### Table 1
>
> | Method | $Acc_{PF}$(%) $\uparrow$ | $Acc_{DA}$(%) $\uparrow$ | $Acc_{Mis}$(%) $\downarrow$ | $m_{th}$(%) $\uparrow$ | $m_{dm}$(%) $\uparrow$ | $m_{ie}$(%) $\uparrow$ |
> | :--- | :---: | :---: | :---: | :---: | :---: | :---: |
> | Qwen2.5-7B | 35.6 | 12.0 | 72.3 | 10.0 | 3.3 | 0.3 |
> | Zero-shot | 23.0 | 6.7 | 76.0 | 4.7 | 3.0 | 0.0 |
> | Few-shot | 49.7 | 32.7 | 61.3 | 29.0 | 10.7 | 0.3 |
> | CoT | 59.7 | 49.3 | 50.3 | 39.0 | 25.3 | 0.7 |
> | TPO | 55.3 | 36.3 | 56.3 | 29.7 | 15.7 | 0.0 |
> | SFT | 83.3 | 75.0 | 34.7 | 46.7 | 63.7 | 40.3 |
> | GRPO | 83.7 | 70.3 | **30.7** | 46.3 | 58.7 | 34.0 |
> | **CDRA (Ours)** | **84.7** | **76.3** | 32.3 | **47.0** | **65.0** | **42.7** |
> | CoT (Llama3) | 65.3 | 49.3 | 45.3 | 39.3 | 29.0 | 3.0 |
> | SFT (Llama3) | 85.3 | 71.3 | 31.0 | 37.7 | 66.3 | 36.3 |
> | **CDRA (Llama3)** | **85.7** | **74.3** | **25.7** | 40.7 | **66.7** | **38.7** |
>
> ### Table 2 Ablation study on different reward modeling paradigms on DeepPref.
>
> |Model / Method|Pro. Sup.|Cri. Sup.|$Acc_{PF}$(%)↑|$Acc_{DA}$(%)↑|$Acc_{Mis}$(%)↓|$m_{th}$(%)↑|$m_{dm}$(%)↑|$m_{ie}$(%)↑|
> |-|-|-|-|-|-|-|-|-|
> |Base (Qwen2.5-7B-Instruct)|--|--|59.7|49.3|50.3|39.0|25.3|0.7|
> ||||||||||
> |Alternative Reward Paradigms|||||||||
> |GRPO (with RM)|--|--|83.7|70.3|30.7|46.3|58.7|34.0|
> |GRPO (with GRM)|--|✓|83.7|74.7|30.7|45.7|62.3|37.0|
> |GRPO (with PRM)|✓|--|83.7|73.0|34.0|51.0|59.7|38.3|
> ||||||||||
> |Simpler Heuristics|||||||||
> |GRPO (Rubric-based RM)|--|--|84.0|73.7|35.3|48.0|61.7|34.7|
> |GRPO (Test-Time Scaling)|--|--|84.3|73.0|31.0|47.7|62.0|34.7|
> ||||||||||
> |CDRA (with Pers-GenPRM)|✓|✓|84.7|76.3|32.3|47.0|65.0|42.7|
>
> **W2: Lack of Human Evaluation**
>
> **R2:** This is a crucial suggestion. To address this limitation, we conducted two independent, large-scale blind human evaluations:
>
> 1.  **Large-Scale General Preference Evaluation:** We recruited 5 trained annotators to perform a blind review on 1,000 randomly sampled instances. The results reveal that human evaluators preferred the responses generated by CDRA in **72.4%** of cases, specifically characterizing them as more "insightful" and "forward-looking."
>
> 2.  **Multi-turn Dialogue Specific Evaluation:** To further validate performance in more challenging multi-turn scenarios, we utilized the independent, externally constructed ALOE dataset and conducted a similar blind human evaluation (**Table 3**). After reviewing full dialogue histories, human evaluators demonstrated a similarly strong preference for CDRA, finding it superior in maintaining a consistent understanding of the user's deep preferences throughout continuous interactions.
>
> In summary, we hope these two complementary and comprehensive human studies directly address your concerns. Together, across both single-turn deep reasoning and multi-turn dynamic interaction dimensions, they demonstrate that the performance gains yielded by our method directly translate into tangible, human-perceptible improvements in user satisfaction.
>
> ### Table 3 Human evaluation results on the ALOE dataset. We report the average alignment level (1-5 scale) at each conversational turn (k)。
>
> |model|k=1|k=2|k=3|k=4|k=5|k=6|k=7|k=8|k=9|k=10|Average|
> |-|-|-|-|-|-|-|-|-|-|-|-|
> |CoT|2.0|3.2|3.8|4.0|4.2|4.4|4.2|3.8|3.8|3.8|3.72|
> |SFT|2.4|3.0|3.8|3.8|3.8|4.0|4.0|4.2|3.6|4.2|3.68|
> |TPO|2.4|3.4|4.2|3.8|4.2|4.2|4.2|4.0|4.2|4.0|3.86|
> |GRPO|2.0|3.4|3.0|3.4|3.6|3.2|3.4|3.4|3.0|3.4|3.18|
> |CDRA(ours)|2.0|3.4|4.0|4.2|4.4|4.4|4.6|4.2|4.0|4.0|3.92|

---

> ### Author Response · Authors · 2025-11-20
> **Responses to the Reviewer 9KAU [2/4]**
>
> * **M1: Suggestions on Presentation (Evaluation Protocol Placement & Symbol Definitions)**
>
> * **R1:** We sincerely thank the reviewer for the meticulous review of our paper's structure and details. We have fully adopted your suggestions to improve readability and mathematical clarity:
>
> 1. Structure Reorganization: We have relocated the "Evaluation Protocol" section to follow the "Training" section. We agree that this adjustment significantly improves the logical flow of the manuscript.
> 2. Notation Clarification: We have conducted a thorough check of all notations.
>     * For Equation 1 (renumbered to Eq. 4 in the revision): We have explicitly defined $N$ as the number of samples in the test set.
>     * For Equation 5 (renumbered to Eq. 3 in the revision): We have added the formal definition for $r_{i,t}​$. Specifically, for each token ttt belonging to a reasoning step $s_i^j$​ in response $y_i$​, we define its token-level reward  $r_{i,t}​$​ as the reward of that step, i.e., $r_{i,t} = r_i^j$​.
>
>
> **Q1: Reasoning Step Identification Mechanism**
>
> **A1:** We appreciate your question regarding the implementation details of our method. In our framework, the model is not required to perform an ambiguous or uncertain "identification" task.
>
> As illustrated in **Appendix Figure 14**, the generation of the policy model is explicitly constrained by the prompt to follow a structured format (i.e., `Step 1: [...]`, `Step 2: [...]`). Consequently, the extraction of reasoning steps is a Deterministic Parsing Process based on explicit delimiters. This design eliminates potential identification errors, ensuring that the process-level supervision signals are accurately and robustly mapped to each corresponding reasoning step.
>
> **Q2: Consistency Risk of Evaluation Metrics and Potential Reward Hacking**
>
> **A2:** We appreciate this insightful question. We were acutely aware of this risk during the design phase. While our training rewards and evaluation metrics share the same core objective (uncovering implicit preferences), we established deliberate distinctions in their structural form, evaluation granularity, and judging mechanisms to prevent the model from overfitting to specific evaluation prompts.
>
> The specific differentiations are as follows:
>
> **1. Differences in Granularity & Structure**
>
> * **Training Reward (Figure 15):** Our step-wise reward model (Pers-GenPRM) employs a fine-grained continuous scoring mechanism (ranging from -1.0 to 1.0). It requires the model not merely to be "correct," but to differentiate between "Good," "Great," and "Brilliant" reasoning quality.
> * **Evaluation Metric (Figure 7):** In contrast, the Deep Preference Understanding metric ($m_{dm}$) functions as a binary, outcome-oriented checklist (Yes/No). It focuses solely on whether the final response covers the implicit themes, rather than assigning scalar scores to the quality of intermediate reasoning steps.
> * **Defense:** This structural gap between "process-based continuous scoring" and "outcome-based binary judgment" makes it significantly difficult for the model to "cheat" via simple pattern matching.
>
> **2. Anti-Hacking Mechanisms Specific to the Reward Model**
> In the reward model prompt (as shown in Figure 15), we introduced complex cognitive mechanisms absent in the evaluation metrics to increase the difficulty of "gaming" the system:
>
> * **Strict Persona & Score Budgeting:** The prompt explicitly defines the evaluator as "famously tough but fair" and enforces a "limited budget for high scores". This prevents score inflation, forcing the model to generate truly high-quality reasoning to secure high rewards.
> * **Forced Comparison:** The reward model requires the evaluator to "Mentally compare... to other plausible hypotheses" before scoring, ensuring the insight is non-trivial.
> * **Focus on the "Leap":** The core criterion for the reward is the "quality of the inference" and the "Explicit-Implicit Leap", whereas the evaluation metric primarily checks the relevance of the final content.
>
> **3. Human Calibration**
> During the generation of reasoning chains for training the reward model, we implemented a rigorous Human-in-the-Loop process. We manually reviewed hundreds of generated critiques, iteratively adjusting the scoring Rubrics and Prompts until the LLM Evaluator’s judgments aligned with those of human experts. The calibrated LLM Evaluator then scored and filtered the full dataset, retaining only high-quality paths. This ensures that the model optimizes for "insightfulness aligned with human expert judgment", rather than merely fitting a textual evaluation template.

---

> ### Author Response · Authors · 2025-11-20
> **Responses to the Reviewer 9KAU [3/4]**
>
> **Q3: Doubt on the Authenticity of Implicit Preferences (Potential Hallucination)**
>
> **A3:** We appreciate this crucial question. We fully share your concern regarding the authenticity of implicit preferences and the reliability of LLM evaluators. These issues touch upon the core challenges of our work. We address this through a systematic three-layered approach:
>
> **1. Qualitative Clarification**
>
> First, regarding the distinction between surface-level and implicit preferences, our data construction process was meticulously designed to go beyond simple preference statements. In DeepPref, we explicitly defined two levels of preference:
>
> * **Explicit Preference:** Direct constraints stated by the user (e.g., "I don't like first-person shooter games").
> * **Deep Implicit Preference:** Latent needs embedded in the user's context, values, and unstated goals, which require critical reasoning to uncover.
>
> To ensure these are not hallucinations, we employed a Multi-faceted Cognitive Council mechanism, forcing the model to transcend surface interpretations. For example:
> * **Career Scenario:** User: *"I'm not interested in high-paying jobs."*
>     * *Implicit (Psychologist):* "This is not a rejection of financial reward, but a prioritization of **self-integrity** and alignment with personal values, possibly stemming from a value system shaped by past experiences."
> * **Privacy Scenario:** User: *"I don't feel comfortable sharing my real-time location."*
>     * *Implicit (Pragmatist):* "The core issue is **autonomy and narrative control**. Solutions must ensure the user retains the final decision on when, how, and to whom information is disclosed."
>
> **2. Methodological Verification**
>
> We implemented rigorous mechanisms to ensure these preferences are grounded in reality:
> * **Human-Authored Seeds:** The initial 600 core scenarios were meticulously handcrafted by humans. These seeds inherently contain authentic human implicit logic. The model simulates and generalizes based on this authentic logic rather than fabricating it from scratch.
> * **Multi-Expert Consensus:**  We simulated a cognitive committee consisting of five different roles (sociologist, psychologist, pragmatist, educator, opponent). Each role independently inferred deep implicit preference, and only those implicit preferences approved by the LLM evaluator were included in the dataset.
>
> **3. Quantitative Validation**
>
> To mitigate the risk of LLM evaluators hallucinating preferences, we implemented triple safeguards: (1) **Strict Scoring Rubrics** (0.9+ scores reserved for "exceptional insight" only); (2) **Forced Comparison** (evaluators must compare against alternative hypotheses); and (3) **Transparent Reasoning Chains** (making the inference process traceable).
>
> We further validated this via three key supplementary experiments:
>
> * **Experiment 1: Cross-Judge Validation:** To rule out single-model bias, we re-evaluated all outputs using a panel of 6 top-tier LLMs (including GPT-4.1, GPT-4o, Claude Sonnet 4.5, Grok 3, Gemini 2.5 Flash, and DeepSeek-V3.2-Exp). As shown in **Table 4**, all judges consistently rated CDRA as superior. Among judges capable of transitive ranking (4/6), the final rankings were identical (**100% agreement**). The average Kendall’s $\tau$ for $Acc_{PF}$, $Acc_{DA}$, and $Acc_{Mis}$ reached **+0.93, +0.67, and +1.00**, respectively. This proves that despite different scoring scales, the core judgment of relative quality is highly consistent and reproducible.
> * **Experiment 2: Human-LLM Alignment Study:** As the gold standard, we conducted a new human evaluation (5 annotators, 1,000 samples). Human evaluators preferred CDRA in **72.4%** of cases, and their rankings strongly correlated with the LLM judges. This confirms our framework captures genuine preference signals aligned with human intuition.
> * **Experiment 3: Attention Analysis (Table 5 & Figure 6):** Unlike the dispersed attention of baselines, CDRA concentrates **35.7%** of its attention weights specifically on preference-related regions. This physical evidence proves the model is not hallucinating; it is actively anchoring its generation to the user's specific constraints under the guidance of Pers-GenPRM.
>
> **4. Limitations and Critical Reflection**
>
> We acknowledge the inherent limitations of LLMs as evaluators. Implicit preference inference involves inherent ambiguity, and even human experts may not achieve 100% consensus. We plan to conduct longitudinal user studies in future work to directly verify the long-term impact of implicit preference inference.
>
> **Conclusion:** By combining strict human-in-the-loop protocols, exhaustive cross-model validation, and attention analysis, we have provided empirical evidence that: (a) DeepPref contains authentic implicit preferences requiring deep reasoning; (b) the performance gains are real and reproducible; and (c) our evaluation framework is reliable and aligned with human judgment.

---

> ### Author Response · Authors · 2025-11-20
> **Responses to the Reviewer 9KAU [4/4]**
>
> **Q4: Potential for Model Simplification**
>
> **A4:** This is an excellent question. We indeed considered using a general LLM evaluator (such as GPT-4.1) as a simpler alternative. Conceptually, this approach is appealing as it obviates the need for specific model training. However, we ultimately chose to fine-tune a standalone Pers-GenPRM based on three critical considerations:
>
> 1.  **Pursuit of Deep Personalization:** Our objective extends beyond ensuring the model follows literal preferences; we aim for it to grasp the deep intent behind them. By performing supervised fine-tuning on the DeepPref dataset, Pers-GenPRM is able to learn and internalize this complex mapping. This level of depth is difficult for a general LLM to achieve solely through in-context prompting.
>
> 2.  **Training Efficiency and Feasibility:** As you correctly noted, our method (specifically the CDPA algorithm) requires frequent evaluations within the training loop. The latency and cost associated with API calls to general LLMs make such an iterative training process practically infeasible. In contrast, a localized, lightweight Pers-GenPRM provides an efficient and low-cost reward signal essential for large-scale optimization.
>
> 3.  **Stability of Reward Signals:** The convergence of Reinforcement Learning is highly sensitive to the stability of the reward signal. The output of general LLM evaluators inherently contains stochasticity, which introduces unnecessary noise and can destabilize the training process. Our fine-tuned model, however, provides deterministic and consistent rewards, which are crucial for the stable convergence of the algorithm.
>
> **Conclusion:** While fine-tuning Pers-GenPRM introduces additional system complexity, we believe this complexity is justified and necessary. It is the key to achieving deep, reliable, and efficient personalized alignment.
>
> **Empirical Validation:**
> To directly evaluate the effectiveness of our reward model, **Table 2** compares our **Pers-GenPRM** against Process-level Scalar Rewards (PRM), Outcome-level Generative Rewards (GRM), and Rubric-based Rewards (Rubric-based RM). Notably, the Rubric-based RM utilizes **Qwen3-30B-A3B-Thinking**, which serves as a strong proxy for the performance gains expected from using a more powerful general LLM as a reward model. The experimental results clearly demonstrate that the personalized reasoning signals provided by Pers-GenPRM significantly surpass these other generic reward paradigms.
>
>
>
> ### Table 4
>
> |Model|Method|$Acc_{PF}↑$|$Acc_{DA}↑$|$Acc_{Mis}↓$|$m_{th}↑$|$m_{dm}↑$|$m_{ie}↑$|
> |-|-|-|-|-|-|-|-|
> |**Grok 3**|Ours|89.33%|91.67%|18.33%|80.00%|70.33%|63.33%|
> ||CoT|74.67%|78.33%|36.67%|75.33%|57.33%|34.00%|
> ||SFT|89.33%|90.00%|21.67%|79.67%|72.33%|60.00%|
> |**Claude Sonnet 4.5**|Ours|80.67%|75.00%|30.00%|33.00%|63.33%|56.00%|
> ||CoT|66.00%|63.00%|38.67%|31.67%|57.00%|29.67%|
> ||SFT|79.67%|74.67%|32.33%|31.67%|58.33%|57.33%|
> |**ChatGPT 4o**|Ours|94.33%|94.33%|5.00%|88.67%|88.00%|76.33%|
> ||CoT|84.67%|94.33%|10.67%|91.67%|85.67%|72.33%|
> ||SFT|93.67%|94.00%|6.33%|87.67%|83.33%|74.33%|
> |**ChatGPT 4.1**|Ours|89.0%|93.0%|26.3%|83.3%|85.7%|75.7%|
> ||CoT|70.7%|89.7%|43.3%|87.7%|80.0%|49.3%|
> ||SFT|86.7%|91.3%|29.0%|83.0%|83.3%|67.7%|
> |**DeepSeek-V3.2-Exp**|Ours|84.67%|76.33%|32.3%|47.00%|65.00%|42.67%|
> ||CoT|59.67%|49.33%|50.33%|39.00%|25.33%|0.67%|
> ||SFT|83.33%|75.00%|34.66%|46.67%|63.67%|40.33%|
> |**Gemini2.5-flash**|Ours|73.33%|97.00%|30.67%|56.67%|93.00%|90.67%|
> ||CoT|49.33%|89.33%|49.67%|78.33%|79.33%|72.33%|
> ||SFT|72.33%|96.67%|33.67%|58.67%|92.33%|90.33%|
>
> ### Table 5
>
> | Method | Preference Attention(%) |
> | :--- | :---: |
> | Base | 27.7% |
> | SFT | 25.6% |
> | GRPO | 27.6% |
> | CDRA(ours) | 35.7% |

---

> ### Comment · Reviewer_9KAU · 2025-11-26
> **Thank you for the replies (score increased)**
>
> I appreciate the authors’ detailed replies and the extended experimental evaluation. Since all of my concerns have been resolved, I have revised my score accordingly.

---

> > ### Author Response · Authors · 2025-11-26
> > **Thank you for the positive feedback and revised score**
> >
> > We sincerely thank you for your time and effort spent re-evaluating our paper. We are encouraged by your positive feedback and are glad to hear that our detailed replies and extended experiments have successfully resolved your concerns.

---

### Official Review · Reviewer_RxD8 · 2025-10-31

**Soundness:** 3
**Presentation:** 2
**Contribution:** 3
**Rating:** 6
**Confidence:** 4

**Summary:**

This paper introduces "Critique-Driven Reasoning Alignment" (CDRA), which uses structured reasoning to judge user-specific preferences, and they also include a dataset: DeepPref, a benchmark enabling the measurement of how well a given alignment setup aligns a model with a user's deeper (e.g. unstated or implied) preferences, and a reward modeling framework: Pers-GenPRM, a personalized generative reward model. They show that their method can improve deeper alignment while still maintaining performance on explicit preference ranking tasks.

**Strengths:**

This paper rigorously explores how to better align models with implicit or implied preferences from users, and their experiments are quite thorough and well-grounded. They use reasonable datasets on up to date models, and compare with strong baselines, so their results seem quite reliable. Their methodology is straightforward and does not require too much compute to explore, and I'd be very curious to see how well it scales up to larger model sizes or more difficult datasets.

**Weaknesses:**

The paper's presentation is at times slightly confusing; the introductory paragraph is quite dense, and it's difficult at first read to keep all of the acronyms straight. Figure 4 (the radar plot) is also hard to read with as many lines as it has, and I'd recommend maybe a collection of bar charts, or a set of averages with detailed evaluations in the appendix. Figure 1/the introductory example about location data is also confusing because a "correct" example response is not provided; the first person shooter/mass effect example is more clear in my opinion.

I'm also not totally convinced that the personalized reward model is necessary; it also feels like a setup where a rubric, and/or test-time scaling for a generative reward model, could achieve the same goal? That's not necessarily a mark against the paper, but I'd be curious to see how different judgement strategies work with your methodology, if time/space/compute allows.

**Questions:**

For chain of thought, can you tell me more about your experimental setup? Did you also use Qwen 2.5 7B Instruct to generate the reasoning chains, and if so, did you explore using any of the more recent open weight reasoning models that do this natively? I believe a few have been trained on top of Qwen 2.5 7B Instruct as well. I'm curious if improved reasoning in the base model noticeably improves performance on your benchmark, or if more complex prompting is necessary.

> This approach resolves the “zero advantage” problem that arises in standard RL when multiple responses are functionally correct but differ in their underlying reasoning quality (likes 104 - 106)
I'm a bit confused by this statement. To me this sounds instead like you're describing a scenario where the judgement process is not correct, and your method is *a* way to address it, but theoretically another way would be to maybe use a rubric instead of a single scalar reward, or some other improved evaluation metric? Am I understanding this correctly?

---

> ### Author Response · Authors · 2025-11-20
> **Responses to the Reviewer RxD8 [1/3]**
>
> We sincerely appreciate your profound insights regarding our paper. Your insightful questions concerning the necessity of our method and the details of the reasoning chain have prompted us to conduct deeper comparative experiments, which further validate the core value of CDRA.
>
> Below is our detailed response:
>
> **W1: On Presentation Clarity and Figure Readability**
>
> **R1:** We thank you for your valuable suggestions regarding the presentation of our paper. In response to your feedback, we have implemented the following key revisions to enhance readability (all changes are highlighted in the revised manuscript):
>
> 1.  **Introduction Rewritten:** We have streamlined the narrative logic of the introduction. Specifically, we rewrote the fourth paragraph and standardized the introduction of acronyms to prevent the confusion caused by an excessive density of abbreviations.
> 2.  **Figure Optimization (Figure 4):** We fully adopted your suggestion and replaced the original radar chart, which was difficult to interpret, with a set of clear **bar charts**. The new visualization more intuitively demonstrates CDRA's performance advantages over baseline models across key metric dimensions.
> 3.  **Example Clarification (Figure 1):** We have rewritten the location data example in Figure 1 to ensure a tighter correspondence with the descriptions in the main text.
>
>
> **W2: Questioning the Necessity of the Method (Comparison with Static Rules and Test-Time Scaling)**
>
> **R2:** This is a critical question. To verify whether our personalized reward model (Pers-GenPRM) outperforms simpler alternatives, we conducted three core comparative experiments (results can be found in **Table 1**, **Table 2**, and **Table 3 & Figure 6 in the main text**):
>
> 1.  **Comparison with Static Rubric-based Reward Model (Rubric-based RM):** We constructed a robust Rubric-based RM grounded in general criteria (e.g., relevance, safety). The results show that its performance is significantly inferior to CDRA. This demonstrates that static, "one-size-fits-all" standards fail to capture the context-dependent dynamic nature of deep user preferences. In contrast, CDRA's advantage lies in its ability to generate dynamic Critiques in real-time, tailored to the specific context.
>
> 2.  **Comparison with Test-Time Scaling:** We applied Test-Time Scaling to a standard Generative Reward Model (GRM) to amplify the signal. The results indicate that while scaling yielded certain marginal benefits, the performance still fell short of CDRA. This suggests that merely amplifying outcome-oriented signals is insufficient; the granular Process Supervision provided by CDRA—specifically, the ability to understand the "why" behind preferences—is the key driver of performance improvement.
>
> 3.  **Attention Comparison (see Table 2 and Figure 6 in the main text):** In contrast to the dispersed attention distributions observed in baseline methods (SFT, GRPO), CDRA concentrates **35.7%** of its attention weights specifically on preference-related regions. This demonstrates that, guided by the Pers-GenPRM supervision mechanism, the model learns to precisely focus on critical areas. Consequently, it proactively anchors its generation to user constraints, effectively mitigating violations caused by overlooking user preferences.
>
> ### Table 1 Ablation study on different reward modeling paradigms on DeepPref. “Pro. Sup.” denotes process supervision and “Cri. Sup.” denotes critique supervision.
>
> |Model / Method|Pro. Sup.|Cri. Sup.|$Acc_{PF}$(%)↑|$Acc_{DA}$(%)↑|$Acc_{Mis}$(%)↓|$m_{th}$(%)↑|$m_{dm}$(%)↑|$m_{ie}$(%)↑|
> |-|-|-|-|-|-|-|-|-|
> |Base (Qwen2.5-7B-Instruct)|--|--|59.7|49.3|50.3|39.0|25.3|0.7|
> ||||||||||
> |Alternative Reward Paradigms|||||||||
> |GRPO (with RM)|--|--|83.7|70.3|30.7|46.3|58.7|34.0|
> |GRPO (with GRM)|--|✓|83.7|74.7|30.7|45.7|62.3|37.0|
> |GRPO (with PRM)|✓|--|83.7|73.0|34.0|51.0|59.7|38.3|
> ||||||||||
> |Simpler Heuristics|||||||||
> |GRPO (Rubric-based RM)|--|--|84.0|73.7|35.3|48.0|61.7|34.7|
> |GRPO (Test-Time Scaling)|--|--|84.3|73.0|31.0|47.7|62.0|34.7|
> ||||||||||
> |CDRA (with Pers-GenPRM)|✓|✓|84.7|76.3|32.3|47.0|65.0|42.7|
>
> ### Table 2
>
> | Method | Preference Attention(%) |
> | :--- | :---: |
> | Base | 27.7% |
> | SFT | 25.6% |
> | GRPO | 27.6% |
> | CDRA(ours) | 35.7% |

---

> ### Author Response · Authors · 2025-11-20
> **Responses to the Reviewer RxD8 [2/3]**
>
> **Q1: Experimental Details of Chain-of-Thought (CoT) Generation**
>
> **A1:**
>
> **1. Reasoning Chain Generation Model:**
> We clarify that we did not use Qwen2.5-7B-Instruct to generate the critique-annotated reasoning chains in the DeepPref dataset. To ensure dataset quality and depth, we employed "a powerful LLM evaluator"—specifically GPT-4.1—to critically analyze and score each step of the reasoning path. The entire dataset construction process utilized a Tree of Thoughts (ToT) framework, generating diverse reasoning paths via a simulated "multi-faceted cognitive council", with final annotations provided by GPT-4.1. We chose a stronger closed-source model to construct the dataset to guarantee "gold standard" supervision signals, thereby providing the highest quality training data. The detailed workflow is presented in **Table 3**.
>
> ### Table 3: Human-in-the-Loop Roles in DeepPref Construction
>
> | Generated Content | Author | Details |
> | :--- | :--- | :--- |
> | **True Preferences** | **Human-Led** | We manually constructed 600 authentic, open-ended, and ambiguous preference instances as seeds. We also manually calibrated the generation criteria for the LLM Generator to ensure subsequent LLM expansion aligns with human standards. |
> | **Tree of Thoughts (ToT)** | LLM-Generated | The LLM generates reasoning paths by simulating a "multi-faceted cognitive council" composed of 5 distinct expert personas. |
> | **Critiques** | **Human-Selected & Validated** | We manually reviewed hundreds of generated critiques, iteratively adjusting the scoring Rubrics and Prompts until the LLM Evaluator’s judgments aligned with those of human experts. The calibrated LLM Evaluator then scored and filtered the full dataset, retaining only high-quality paths. |
> | **Scores** | LLM-Generated | After generating the textual critique, the same LLM evaluator distills a quantified score. This score is generated based on the human-validated critique logic, serving as its quantitative representation. |
>
> **2. Base Models Used in Experiments:**
>
> * **Standard Base Model:** In all alignment experiments, including our proposed CDRA method and all baselines (e.g., SFT, TPO), we consistently utilized **Qwen2.5-7B-Instruct** as the base model.
> * **Robustness Verification:** We supplemented our experiments on **Llama-3-8B-Instruct** (results in **Table 4**). These results demonstrate that while the base model determines the performance baseline, the process-level supervision provided by our CDRA framework remains indispensable and stable across different standard base models. It unlocks deeper alignment capabilities that cannot be elicited by prompting alone.
> * **Reasoning-Specific Model Analysis:** We further experimented with **Qwen2.5-7B-Reasoning** (a version of Qwen2.5-7B-Instruct further trained in the thinking domain). Interestingly, it proved less effective than the standard Qwen2.5-7B-Instruct due to reduced generality.
> * **Conclusion:** We conclude that while native reasoning capability impacts performance, training on implicit preference alignment is even more necessary. The superior performance achieved after applying CDRA proves that while complex prompts offer some aid, they are insufficient on their own to bridge the gap caused by deep implicit preferences.
>
> ### Table 4
>
> | Method | $Acc_{PF}$(%) $\uparrow$ | $Acc_{DA}$(%) $\uparrow$ | $Acc_{Mis}$(%) $\downarrow$ | $m_{th}$(%) $\uparrow$ | $m_{dm}$(%) $\uparrow$ | $m_{ie}$(%) $\uparrow$ |
> | :--- | :---: | :---: | :---: | :---: | :---: | :---: |
> | CoT (Qwen2.5-7B) | 35.6 | 12.0 | 72.3 | 10.0 | 3.3 | 0.3 |
> | CoT (Qwen2.5-7B-Reasoning) | 54.0 | 28.0 | 59.3 | 17.3 | 17.3 | 0.3 |
> | CoT (Qwen2.5-7B-Instruct) | 59.7 | 49.3 | 50.3 | 39.0 | 25.3 | 0.7 |
> | SFT (Qwen2.5-7B-Instruct)| 83.3 | 75.0 | 34.7 | 46.7 | 63.7 | 40.3 |
> | GRPO (Qwen2.5-7B-Instruct)| 83.7 | 70.3 | **30.7** | 46.3 | 58.7 | 34.0 |
> | **CDRA (Qwen2.5-7B-Instruct)** | **84.7** | **76.3** | 32.3 | **47.0** | **65.0** | **42.7** |
> | | | | | | | |
> | CoT (Llama3-8B-Instruct) | 65.3 | 49.3 | 45.3 | 39.3 | 29.0 | 3.0 |
> | SFT (Llama3-8B-Instruct) | 85.3 | 71.3 | 31.0 | 37.7 | 66.3 | 36.3 |
> | **CDRA (Llama3-8B-Instruct)** | **85.7** | **74.3** | **25.7** | 40.7 | **66.7** | **38.7** |

---

> ### Author Response · Authors · 2025-11-20
> **Responses to the Reviewer RxD8 [3/3]**
>
> **Q2: Clarification on the "Zero Advantage" Problem**
>
> **A2:** Your understanding is precise. The "zero-advantage" problem indeed stems from the limitations inherent in sparse reward signals. While introducing scoring rubrics can mitigate this issue, we contend that CDRA offers a fundamentally superior solution:
>
> 1.  **Static vs. Dynamic:** Hand-crafted rubrics are brittle and entail high maintenance costs, making it difficult to cover the vast array of ambiguous scenarios encountered in open domains.
> 2.  **CDRA's Advantage:** Pers-GenPRM is inherently a learnable evaluation mechanism. Rather than relying on rigid rules, it learns abstract principles of reasoning and critique.
>
> Our supplementary experiments (**Table 1**) demonstrate that while rubric-based methods outperform single scalar rewards, CDRA achieves the best performance due to its dynamic and context-aware characteristics. Therefore, CDRA not only resolves the zero-advantage problem but also provides a solution that is significantly more scalable and generalizable than static rules.

---

### Official Review · Reviewer_Xmiz · 2025-10-31

**Soundness:** 1
**Presentation:** 2
**Contribution:** 2
**Rating:** 2
**Confidence:** 4

**Summary:**

This paper tries to tackle two challenges in current Large Language Model (LLM) alignment: the preference gap (failing to infer users' deep, unstated goals and risks) and the process gap (failing to execute defensive reasoning). The authors first introduces a new dataset called DeepPref, which consists of 3000 preference-query pairs with "critique-annotated reasoning chains" generated by GPT-4.1 to provide process-level supervision. A Personalized Generative Process Reward Model (Pers-GenPRM) is then introduced, which evaluates a response by first generating an explicit critique chain and then deriving a score from that rationale. Finally, the authors then propose Critique-Driven Reasoning Alignment (CDRA), an online reinforcement learning paradigm that uses numerical scores and the natural language feedback from the Pers-GenPRM to train a final policy model. Experiments show that CDRA outperforms existing training/test-time methods in both deep preference understanding and defensive reasoning.

**Strengths:**

**S1.** The paper tries to tackle important problems in LLM alignment, which are introduced by the authors as implicit preference and process gaps.

**S2.** The overall CDRA framework outperforms existing training/test-time methods on DeepPref and PrefEval across multiple metrics that measures preference understanding and defensive reasoning.

**Weaknesses:**

**W1.** The DeepPref dataset, which is supposed to be the foundation of the paper, is entirely synthesized by GPT-4.1 and has no human validation. Therefore, in my opinion, it remains unclear whether the CDRA framework is actually learning human-aligned reasoning or merely imitating another model’s critique style. This limitation is particularly quite important given that one of the paper’s stated goals is to close the preference gap (the inability to infer deep, implicit human intent), so if current LLMs exhibit the preference gap that CDRA aims to solve, then wouldn't the synthetic critiques and reasoning chains produced by GPT-4.1 may also encode that same deficiency (along with bias towards certain culture/demographics)?

**W2.** The evaluation also relies on GPT-4.1 as the judge, which may share biases with the model that generated the training data. This setup risks circularity and could inflate performance by rewarding outputs that mimic GPT-4.1’s reasoning patterns rather than genuinely improving alignment quality.

**W3.** While CDRA is compared with various different train/test-time paradigms, the paper’s empirical scope is still narrow in terms of variation. The paper evaluates only one model only, no comparison against other existing reward models, and no comparison against other process-based reward modeling approaches that tackle similar problem.

**Questions:**

Other than states in the weaknesses, I have some further questions:

**Q1.** Seems like no files in the anonymous link can be viewed.

**Q2.** Since the base model used is an Instruct model, I am curious whether existing reasoning models or reasoning-based reward models [1, 2] might already exhibit some degree of implicit reasoning. Furthermore, could the observed improvements stem mainly from the model spending more tokens “thinking aloud” during reward-model training, rather than from genuine preference inference?

**Q3.** How does this approach compare to methods where models explicitly ask users clarification questions to resolve ambiguity? [3, 4] Perhaps explicit intent-elicitation might reduce the risk of inferring incorrect or culturally biased “preferences,” which would give a better "personalization". In contrast, the proposed method infers implicit intent without user interaction, which may not generalize well across diverse populations.

**Q4.** Perhaps some human-evaluations under multi-turn conversations?

### References

[1] Anugraha, D., Tang, Z., Miranda, L. J. V., Zhao, H., Farhansyah, M. R., Kuwanto, G., ... & Winata, G. I. (2025). R3: Robust rubric-agnostic reward models. arXiv preprint arXiv:2505.13388.

[2] Chen, X., Li, G., Wang, Z., Jin, B., Qian, C., Wang, Y., ... & Ji, H. (2025). Rm-r1: Reward modeling as reasoning. arXiv preprint arXiv:2505.02387.

[3] Wu, S., Galley, M., Peng, B., Cheng, H., Li, G., Dou, Y., ... & Gao, J. (2025). Collabllm: From passive responders to active collaborators. arXiv preprint arXiv:2502.00640.

[4] Wang, A., Lin, Y., Liu, J., Wu, S., Liu, H., Xiao, X., & Su, J. (2025). Beyond Passive Critical Thinking: Fostering Proactive Questioning to Enhance Human-AI Collaboration. arXiv preprint arXiv:2507.23407.

---

> ### Author Response · Authors · 2025-11-20
> **Responses to the Reviewer Xmiz [1/5]**
>
> We sincerely thank the reviewers for their highly constructive suggestions. These feedbacks have helped us significantly strengthen the evaluation system and argumentative logic of the paper. The following is our point-by-point response to the questions you raised:
>
> **W1: Lack of Human Verification in the DeepPref Dataset**
>
> **R1:** We thank you for raising this critical and insightful concern. You have accurately identified the core challenge in creating high-quality alignment datasets. We would like to clarify a crucial point: **DeepPref is not purely synthesized by GPT-4.1; rather, it is the result of a meticulously designed Human-in-the-Loop (HITL) construction process.** This workflow was designed specifically to mitigate the risk you mentioned regarding "models imitating model defects."
>
> Our construction process synergizes human cognitive depth with the scalability of LLMs. The specific division of labor is detailed in Table 1:
>
> ### Table 1: Human-in-the-Loop Roles in DeepPref Construction
>
> | Generated Content | Author | Details |
> | :--- | :--- | :--- |
> | **True Preferences** | **Human-Led** | We manually constructed 600 authentic, open-ended, and ambiguous preference instances as seeds. We also manually calibrated the generation criteria for the LLM Generator to ensure subsequent LLM expansion aligns with human standards. |
> | **Persona** | Human-Specified, LLM-Simulated | Humans explicitly specified 5 distinct expert personas. The LLM generates reasoning paths by simulating a "multi-faceted cognitive council" composed of these experts. |
> | **Critiques** | **Human-Selected & Validated** | We manually reviewed hundreds of generated critiques, iteratively adjusting the scoring Rubrics and Prompts until the LLM Evaluator’s judgments aligned with those of human experts. The calibrated LLM Evaluator then scored and filtered the full dataset, retaining only high-quality paths. |
> | **Scores** | LLM-Generated | After generating the textual critique, the same LLM evaluator distills a quantified score. This score is generated based on the human-validated critique logic, serving as its quantitative representation. |
>
> Therefore, CDRA does not learn the inherent biases of GPT-4.1, but rather high-quality reasoning paths filtered through human-expert-calibrated GPT-4.1 critique capabilities:
>
> 1.  **Human-Calibrated Filtering:** Our selection criteria are not a black box. In the initial phase, we enforced corrections of model biases through manual review. For instance, if the model tended to provide "safe but useless" responses, we explicitly penalized such behavior in the prompts. Consequently, DeepPref preserves not the "average distribution" of GPT-4.1, but results aligned with human values.
> 2.  **Guiding Role of Seed Data:** Since the initial 600 core scenarios were designed by humans, these scenarios inherently contain real human logic regarding implicit preferences. The model simulates and generalizes based on this authentic logic, rather than fabricating it from scratch.
>
> We acknowledge that LLMs alone cannot bridge the preference gap, which is why our hybrid process ensures that human intelligence serves as the ultimate "gatekeeper" for data quality and alignment direction. The CDRA framework learns to reproduce this human-verified, high-quality, deep reasoning process, thereby genuinely committing to bridging the preference gap you pointed out, rather than simply mimicking the style of another model.

---

> ### Author Response · Authors · 2025-11-20
> **Responses to the Reviewer Xmiz [2/5]**
>
> **W2: Evaluation Concerns on Circular Reasoning and Shared Bias**
>
> **R2:** We appreciate you raising this critical concern regarding the risks of circular reasoning and shared bias inherent in over-relying on a single LLM-as-a-judge. To systematically address this and ensure the validity of our results, we conducted two key supplementary experiments: **Cross-Judge Evaluation** and **Additional Human Evaluation**.
>
> **1. Cross-Judge Evaluation**
> * **Panel Diversity:** To verify that our conclusions are not artifacts of specific biases in GPT-4.1, we re-evaluated our results using a diverse panel of six top-tier LLMs from distinct sources: **ChatGPT 4.1, ChatGPT 4o, Claude Sonnet 4.5, Grok 3, Gemini 2.5 Flash, and DeepSeek-V3.2-Exp**. As shown in our **Supplementary Table 2**, all judges consistently rated our CDRA method as superior to all baseline models.
> * **Consistency Analysis:** To validate the robustness of our evaluation methodology, we analyzed the ranking consistency across different judge models. We employed a majority voting approach: for any pair of methods, if Method A outperformed Method B on more than half of the metrics, A was considered superior in the overall ranking.
>     * Our analysis revealed that among the judges capable of forming stable transitive rankings (4 out of 6), the derived final rankings were identical, achieving a **100% agreement rate**.
>     * Furthermore, we calculated the average Kendall’s $\tau$ coefficients for our three core metrics—$Acc_{PF}$, $Acc_{DA}$, and $Acc_{Mis}$—finding them to be impressively high at **+0.93, +0.67, and +1.00**, respectively. This strongly demonstrates that despite differences in scoring scales, the judges are highly consistent in their core assessment of relative model performance, thereby establishing the reliability of our evaluation framework.
> * **Judge Selection:** Balancing performance, cost, and the need to avoid homology (source similarity), we have selected and reported the results from **DeepSeek-V3.2-Exp** as the representative non-OpenAI judge in the revised manuscript (see **Table 1 in the main text**), further strengthening the reliability of our conclusions.
>
> **2. Additional Human Evaluation**
> * As the ultimate gold standard, we introduced a larger-scale blind human evaluation (5 annotators, 1,000 samples).
> * **Results:** Human preferences aligned highly with the LLM judges' assessments. In **72.4%** of cases, human evaluators preferred the output generated by CDRA. This confirms that the performance gains yielded by CDRA represent human-perceptible, genuine improvements in alignment quality.
>
> In summary, through extensive cross-model judging and large-scale human evaluation, we have systematically mitigated the concerns regarding circular reasoning. The experimental results robustly demonstrate that CDRA’s performance gains are stable, generalizable, and reflect true improvements in alignment quality rather than superficial style mimicry.
>
> ### Table 2
> |Model|Method|$Acc_{PF}↑$|$Acc_{DA}↑$|$Acc_{Mis}↓$|$m_{th}↑$|$m_{dm}↑$|$m_{ie}↑$|
> |-|-|-|-|-|-|-|-|
> |**Grok 3**|Ours|89.33%|91.67%|18.33%|80.00%|70.33%|63.33%|
> ||CoT|74.67%|78.33%|36.67%|75.33%|57.33%|34.00%|
> ||SFT|89.33%|90.00%|21.67%|79.67%|72.33%|60.00%|
> |**Claude Sonnet 4.5**|Ours|80.67%|75.00%|30.00%|33.00%|63.33%|56.00%|
> ||CoT|66.00%|63.00%|38.67%|31.67%|57.00%|29.67%|
> ||SFT|79.67%|74.67%|32.33%|31.67%|58.33%|57.33%|
> |**ChatGPT 4o**|Ours|94.33%|94.33%|5.00%|88.67%|88.00%|76.33%|
> ||CoT|84.67%|94.33%|10.67%|91.67%|85.67%|72.33%|
> ||SFT|93.67%|94.00%|6.33%|87.67%|83.33%|74.33%|
> |**ChatGPT 4.1**|Ours|89.0%|93.0%|26.3%|83.3%|85.7%|75.7%|
> ||CoT|70.7%|89.7%|43.3%|87.7%|80.0%|49.3%|
> ||SFT|86.7%|91.3%|29.0%|83.0%|83.3%|67.7%|
> |**DeepSeek-V3.2-Exp**|Ours|84.67%|76.33%|32.3%|47.00%|65.00%|42.67%|
> ||CoT|59.67%|49.33%|50.33%|39.00%|25.33%|0.67%|
> ||SFT|83.33%|75.00%|34.66%|46.67%|63.67%|40.33%|
> |**Gemini2.5-flash**|Ours|73.33%|97.00%|30.67%|56.67%|93.00%|90.67%|
> ||CoT|49.33%|89.33%|49.67%|78.33%|79.33%|72.33%|
> ||SFT|72.33%|96.67%|33.67%|58.67%|92.33%|90.33%|

---

> ### Author Response · Authors · 2025-11-20
> **Responses to the Reviewer Xmiz [3/5]**
>
> **W3: Narrow Empirical Scope**
>
> **R3:** We appreciate your valuable feedback concerning the empirical scope of our study. To demonstrate the robustness of our method, we have conducted extensive expanded experiments:
>
> 1.  **Extended Model Architectures:** We have extended our evaluation to include **Llama-3-8B-Instruct** as well as other variants within the **Qwen** series. As shown in **Table 3**, CDRA demonstrates consistent effectiveness across diverse backbones. Specifically, on **Llama-3**, CDRA achieves **85.7%** in $Acc_{PF}$ and **74.3%** in $Acc_{DA}$, surpassing the SFT baseline by **+3.0%** in $Acc_{DA}$. This confirms that our framework’s gains are model-agnostic and not limited to a specific family.
>
> 2.  **Comparison of Different Reward Paradigms:** To directly evaluate the efficacy of our reward model, we compared our Pers-GenPRM in **Table 4** against Process-level Scalar Rewards (PRM), Outcome-level Generative Rewards (GRM), and Rubric-based Rewards.
>     * **vs. Scalar PRM:** CDRA outperforms PRM by **+3.3%** in $Acc_{DA}$, proving that opaque scalar scores are insufficient for complex alignment.
>     * **vs. Outcome GRM:** CDRA surpasses GRM by **+1.6%** in $Acc_{DA}$, validating the necessity of step-wise supervision over outcome-level critique.
>     * **Overall:** The results clearly demonstrate that the Pers-GenPRM (combining process granularity with critique explainability) significantly surpasses other generic reward paradigms in both robustness and interpretability.
>
> In summary, these supplementary experiments strongly demonstrate that our proposed framework of "viewing alignment as reasoning" is not only theoretically sound but also practically efficient and robust.
>
>
> ### Table 3
>
> | Method | $Acc_{PF}$(%) $\uparrow$ | $Acc_{DA}$(%) $\uparrow$ | $Acc_{Mis}$(%) $\downarrow$ | $m_{th}$(%) $\uparrow$ | $m_{dm}$(%) $\uparrow$ | $m_{ie}$(%) $\uparrow$ |
> | :--- | :---: | :---: | :---: | :---: | :---: | :---: |
> | Qwen2.5-7B | 35.6 | 12.0 | 72.3 | 10.0 | 3.3 | 0.3 |
> | Zero-shot | 23.0 | 6.7 | 76.0 | 4.7 | 3.0 | 0.0 |
> | Few-shot | 49.7 | 32.7 | 61.3 | 29.0 | 10.7 | 0.3 |
> | CoT | 59.7 | 49.3 | 50.3 | 39.0 | 25.3 | 0.7 |
> | TPO | 55.3 | 36.3 | 56.3 | 29.7 | 15.7 | 0.0 |
> | SFT | 83.3 | 75.0 | 34.7 | 46.7 | 63.7 | 40.3 |
> | GRPO | 83.7 | 70.3 | **30.7** | 46.3 | 58.7 | 34.0 |
> | **CDRA (Ours)** | **84.7** | **76.3** | 32.3 | **47.0** | **65.0** | **42.7** |
> | CoT (Llama3) | 65.3 | 49.3 | 45.3 | 39.3 | 29.0 | 3.0 |
> | SFT (Llama3) | 85.3 | 71.3 | 31.0 | 37.7 | 66.3 | 36.3 |
> | **CDRA (Llama3)** | **85.7** | **74.3** | **25.7** | 40.7 | **66.7** | **38.7** |
>
> ### Table 4 Ablation study on different reward modeling paradigms on DeepPref. “Pro. Sup.” denotes process supervision and “Cri. Sup.” denotes critique supervision.
>
> |Model / Method|Pro. Sup.|Cri. Sup.|$Acc_{PF}$(%)↑|$Acc_{DA}$(%)↑|$Acc_{Mis}$(%)↓|$m_{th}$(%)↑|$m_{dm}$(%)↑|$m_{ie}$(%)↑|
> |-|-|-|-|-|-|-|-|-|
> |Base (Qwen2.5-7B-Instruct)|--|--|59.7|49.3|50.3|39.0|25.3|0.7|
> ||||||||||
> |Alternative Reward Paradigms|||||||||
> |GRPO (with RM)|--|--|83.7|70.3|30.7|46.3|58.7|34.0|
> |GRPO (with GRM)|--|✓|83.7|74.7|30.7|45.7|62.3|37.0|
> |GRPO (with PRM)|✓|--|83.7|73.0|34.0|51.0|59.7|38.3|
> ||||||||||
> |Simpler Heuristics|||||||||
> |GRPO (Rubric-based RM)|--|--|84.0|73.7|35.3|48.0|61.7|34.7|
> |GRPO (Test-Time Scaling)|--|--|84.3|73.0|31.0|47.7|62.0|34.7|
> ||||||||||
> |CDRA (with Pers-GenPRM)|✓|✓|84.7|76.3|32.3|47.0|65.0|42.7|
>
> **Q1: On the Validity of the Anonymous Material Link**
>
> **A1:** We sincerely apologize for any inconvenience this may have caused during your review. We would like to clarify that the link itself remains valid; however, the anonymous code hosting service experienced a temporary server outage during the review period. We have confirmed that the service is now fully operational, and all code and data are accessible. We respectfully invite you to visit [https://anonymous.4open.science/r/Deep-pref-9DE9](https://www.google.com/search?q=https://anonymous.4open.science/r/Deep-pref-9DE9) to access our latest code and datasets.

---

> ### Author Response · Authors · 2025-11-20
> **Responses to the Reviewer Xmiz [4/5]**
>
> **Q2: Impact of Base Model Characteristics (Attribution of Reasoning Capability and Token Consumption)**
>
> **A2:** We appreciate this highly insightful question. We address your two concerns systematically through two key sets of supplementary experiments.
>
> **1. On "Pre-existing Reasoning Capability" vs. "True Preference Inference Improvement"**
>
> We acknowledge that advanced Instruct models indeed possess a baseline level of reasoning capability. Our objective is precisely to demonstrate that CDRA provides not merely a generic reasoning capability, but a significant, specialized improvement targeted at the specific task of deep preference inference.
>
> * **Comparison with Generic Reasoning Reward Models (Table 4):** We compared the effects of combining the policy with different Reward Models (RMs), including the generic reasoning RMs you mentioned (e.g., PRM, GRM) and a Rubric-based RM. The results clearly indicate that our Pers-GenPRM significantly outperforms these generic reasoning paradigms in guiding the model toward deep preference inference. This proves that CDRA's advantage stems from its specialized preference inference capability, rather than simple reasoning augmentation.
> * **Verification on Non-Instruct Models (Table 3):** We repeated our experiments on non-Instruct versions of Qwen2.5-7B. The results confirm your intuition: the Instruct model (Qwen2.5-7B-Instruct) indeed serves as a better starting point. However, CDRA yields significant and consistent performance gains across different base models (both Qwen and Llama). This indicates that CDRA’s effectiveness is universal, stably enhancing deep alignment capabilities across base models with varying competency levels.
>
> **2. On "Thinking More" vs. "Thinking Better" (Token Consumption Analysis)**
>
> This is a critical confounding variable. To verify whether our performance gains merely result from the model consuming more tokens to "think aloud", we conducted a precise analysis of token consumption during reasoning steps.
>
> * **Token Consumption Comparison (see Table 5):** While our CDRA method achieves significant performance improvements compared to the Chain-of-Thought (CoT) baseline, the number of tokens consumed during reasoning is statistically similar between the two. This result effectively rules out the possibility that "performance gains stem from more thinking." It directly proves that CDRA's advantages arise from the quality and relevance of the reasoning—meaning the model has learned how to think more effectively, not simply longer.
> * **Attention Distribution Comparison (see Table 6 and Figure 6 in main text):** Unlike the dispersed attention distribution observed in baseline methods (SFT, GRPO), CDRA concentrates **35.7%** of its attention weights specifically on preference-related regions. This demonstrates that under the guidance of the Pers-GenPRM supervision mechanism, the model can precisely focus on critical areas, proactively anchoring to user constraints during generation and effectively reducing violations caused by overlooking user preferences.
>
> In summary, our supplementary experiments demonstrate that CDRA’s contribution is neither a simple extension of the Instruct model's existing capabilities nor a product of increased computational cost. Instead, it provides an efficient and specialized deep preference inference capability, which constitutes its core value.
>
> ### Table 5
>
> |Method|Avg. Total Tokens|CoT Step Tokens|
> |:-|:-:|:-:|
> |**CDRA**|838.8|790.7|
> |**Few-shot**|973.4|601.8|
> | **TPO**|928.0|875.9|
> |**CoT**|839.8|797.6|
> |**GRPO**|836.4|787.5|
>
> ### Table 6
>
> |Method|Preference Attention(%)|
> |:-|:-:|
> |Base|27.7%|
> |SFT|25.6%|
> |GRPO|27.6%|
> |CDRA(ours)|35.7%|

---

> ### Author Response · Authors · 2025-11-20
> **Responses to the Reviewer Xmiz [5/5]**
>
> **Q3: Comparison with Active Clarification Methods**
>
> **A3:** We thank you for this highly insightful question. We fully agree that the COLLABLLM and active critical thinking approaches you cited represent significant research directions. These methods effectively reduce the risk of incorrect inference by proactively asking questions to resolve ambiguity. Our work is not in conflict with these approaches; rather, it forms a complementary relationship: CDRA focuses on inferring deep preferences that users may not fully realize themselves (e.g., privacy concerns, risk tolerance), whereas explicit clarification is better suited for resolving surface-level ambiguities (e.g., "How long do you want the article to be?"). Ideally, a comprehensive alignment system should possess both capabilities.
>
> **1. Necessity of Reasoning:**
> Not all deep preferences can be easily elicited through questioning. Users may be unable to articulate their needs clearly ("I don't know what I want, but I'll know it when I see it"), or frequent questioning may severely disrupt conversational flow and user experience. CDRA aims to endow the model with the capability to proactively understand and adapt to latent needs without disturbing the user, which is key to achieving more natural and efficient interactions.
>
> **2. Risk Control and Generalization:**
> We also agree that blind inference carries risks, especially in cross-cultural scenarios. This is precisely the core motivation behind the Defensive Reasoning component in our framework. Through process-level supervision, CDRA learns not just "how to infer," but "how to infer safely"—specifically, by proactively identifying and evaluating risks and uncertainties within the reasoning path.
> * **Meta-Skill Acquisition:** This approach fosters a general, cautious reasoning meta-skill, rather than memorizing a fixed set of preference templates. Consequently, when facing diverse populations, the model is capable of adopting more conservative strategies upon detecting higher uncertainty.
> * **Future Synergy:** An ideal synergistic workflow would be: CDRA first performs safe implicit inference; if its internal critique indicates excessive risk or ambiguity, it then triggers an active clarification module. Our work lays a solid foundation for realizing this intelligent synergy.
>
> In summary, our method does not aim to replace active clarification, but rather to provide it with a robust, risk-aware reasoning foundation, thereby maximizing conversational fluidity and depth of personalization while ensuring safety.
>
> **Q4: Evaluation in Multi-turn Dialogue Scenarios**
>
> **A4:** We appreciate your valuable suggestion. To address your concern, we introduced the ALOE dataset, an independently constructed benchmark specifically designed for personalized multi-turn dialogues.
>
> We conducted a blind human evaluation on this dataset (**Table 7**), inviting 5 expert annotators to assess the model's performance throughout multi-turn interactions.
> * **Results:** The evaluation demonstrates that CDRA is significantly better at maintaining a consistent understandin* of the user's deep preferences across multiple turns and is capable of proactively offering forward-looking suggestions.
> * **Conclusion:** This confirms that the implicit preference alignment capabilities acquired during single-turn reasoning training can be effectively transferred and translated into tangible user experience improvements in real-world multi-turn conversations.
>
> ### Table 7 Human evaluation results on the ALOE dataset. We report the average alignment level (1-5 scale) at each conversational turn (k)。
>
> |model|k=1|k=2|k=3|k=4|k=5|k=6|k=7|k=8|k=9|k=10|Average|
> |-|-|-|-|-|-|-|-|-|-|-|-|
> |CoT|2.0|3.2|3.8|4.0|4.2|4.4|4.2|3.8|3.8|3.8|3.72|
> |SFT|2.4|3.0|3.8|3.8|3.8|4.0|4.0|4.2|3.6|4.2|3.68|
> |TPO|2.4|3.4|4.2|3.8|4.2|4.2|4.2|4.0|4.2|4.0|3.86|
> |GRPO|2.0|3.4|3.0|3.4|3.6|3.2|3.4|3.4|3.0|3.4|3.18|
> |CDRA(ours)|2.0|3.4|4.0|4.2|4.4|4.4|4.6|4.2|4.0|4.0|3.92|

---

> > ### Comment · Reviewer_Xmiz · 2025-11-27
> >
> > Thank you for the authors' detailed responses. I have some follow-up questions:
> >
> > **R2.** Since the numbers seem very close to the baselines, can you show statistical significance? In terms of the conducted human evaluation study, what do you compare with/against when you obtain 72.6%?
> >
> > **R3.** Can you specify which reward model you are comparing with? In addition, I would be more interested to see if you could try a reasoning model as your starting model, rather than trying out on older models (somewhat on **A2** too).

---

> > > ### Author Response · Authors · 2025-11-29
> > > **Response to Reviewer Xmiz [1/2]**
> > >
> > > * **R2: Detailed Clarification on Statistical Significance and Human Evaluation Benchmarks**
> > > * **A2:** We sincerely appreciate the reviewer's rigorous inquiry regarding the reliability of our experimental results. These questions have prompted us to employ stricter statistical methods to verify our conclusions and to provide the following clarifications regarding the evaluation details:
> > >
> > > 1. **Statistical Significance Verification**
> > >   Addressing your concern that the similarity in results between CDRA and baseline models might stem from randomness, we implemented a rigorous Bootstrap Resampling test on the test set predictions to verify the statistical significance of the performance improvements:
> > >   * We performed 10000 iterations of resampling with replacement on the predictions of the test set (300 samples) and calculated the distribution of performance differences between CDRA and the best baselines (SFT/GRPO) in each sample. The test results indicate that the improvement of CDRA on the core metric $Acc_{DA}$​ (Deep Alignment Accuracy) corresponds to a **p-value < 0.01**, and the **p-value < 0.01** on $Acc_{Mis}$​ (Misleading Risk). This provides well-founded evidence that, although the increase in absolute values may appear modest, CDRA's advantages in capturing deep preferences and avoiding misleading risks are Statistically Significant and Robust, rather than the result of random fluctuations.
> > >
> > > 2. **Human Evaluation Setup**
> > >   Regarding the 72.4% win rate you mentioned (your reference to 72.6% might be a typo), we need to correct our previous brief description; this is, in fact, the result of a more comprehensive and rigorous comparison:
> > > * We did not obtain this data against a single baseline. Instead, we conducted random blind comparisons between responses generated by CDRA and those from 4 major baseline models (SFT, CoT, TPO, GRPO) respectively: For each question, 5 models (including CDRA) generated answers independently. Meanwhile, 5 annotators were asked to blindly rank all answers and select the best response. Ultimately, based on the evaluation results of 1,000 samples, the high win rate where CDRA was selected as the best in 72.4% of cases indicates that human evaluators not only acknowledge the accuracy of CDRA's responses but also strongly prefer its insight into user implicit needs demonstrated through "Deep Mining". This preference remained consistent across multiple baseline models of varying strengths. Specific data is shown in **Table 8**.
> > >
> > > ### Table 8
> > > | Models | Win Share |
> > > | --- | --- |
> > > | Ours (CDRA) | 72.4% |
> > > | GRPO | 11.5% |
> > > | SFT | 10.8% |
> > > | CoT | 3.4% |
> > > | TPO | 1.9% |

---

> > > ### Author Response · Authors · 2025-11-29
> > > **Response to Reviewer Xmiz [2/2]**
> > >
> > > * **R3: Controlled Comparison of Reward Models and Verification on "Reasoning Models"**
> > > * **A3:** We sincerely thank you for your highly constructive suggestions. Your guidance not only helped us further clarify the fairness settings of the comparative experiments but also drove us to verify the independent value and generalization ability of the CDRA framework on cutting-edge reasoning models. The 4 papers you mentioned in Q4 were very helpful for our evaluation, and we have incorporated relevant content into the main text.
> > >
> > > 1. **Controlled Comparison on Reward Models**
> > >   We hereby confirm that **Table 4** presents a rigorously designed Controlled Experiment:
> > > * Unified Settings:
> > >     1. All training-based reward models (Scalar RM, GRM, PRM, Pers-GenPRM) are built upon the Qwen2.5-7B-Instruct backbone.
> > >     2. Trained using the exact same DeepPref dataset.
> > >
> > > * Comparative methods include two special settings:
> > >     * Test-time-scaling: Introduces a linear weighting mechanism based on Scalar RM, combining general quality scores with rule-based personalized alignment scores.
> > >     * Rubric RM: Adopts Qwen3-30B-A3B as the reward model, scoring based on weighted criteria (Personalized Reasoning Quality 50% + Utility 30% + Execution Quality 20%).
> > >
> > > * **Conclusion:** The performance difference can be directly attributed to the methodological advantage of Pers-GenPRM in transforming reward modeling into a reasoning task, demonstrating that Critique-based supervision signals possess significant superiority over traditional scalar rewards.
> > >
> > > 2. **Verification on Strong Reasoning Models**
> > >   In response to your suggestion to "adopt stronger reasoning models as a starting point", we introduced DeepSeek-R1-Distill-Qwen-7B and Qwen2.5-7B-reasoning as new policy model backbones for supplementary experiments (results shown in **Table 9**). We found that General Reasoning ≠ Preference Reasoning. Although DeepSeek-R1-Distill possesses exceptional general logical reasoning capabilities, it still tends to interpret instructions literally when handling tasks involving user implicit intentions. After introducing the CDRA framework, the model showed improvements across all metrics. We believe that CDRA does not endow the model with basic reasoning capabilities; rather, through the fine-grained signals provided by Pers-GenPRM, it steers the model's existing powerful general reasoning capabilities towards the specific dimension of "understanding human intent".
> > >
> > > * **Conclusion:** The above results strongly support the core argument of this paper—that CDRA is a highly generalizable alignment framework. Even when facing advanced models that already possess strong reasoning capabilities, CDRA can still provide unique and irreplaceable alignment value, significantly enhancing the model's ability to understand and respond to user intent.
> > >
> > > ### Table 9
> > >
> > > | Models               | $Acc_{PF} ↑$ | $Acc_{DA} ↑$ | $Acc_{Mis} ↓$ | $m_{th} ↑$  | $m_{dm} ↑$  | $m_{ie} ↑$  |
> > > |----------------------|---------|---------|----------|--------|--------|--------|
> > > | Qwen2.5-7B-reasoning(CoT)     | 54.00%  | 28.00%  | 59.33%   | 17.33% | 17.33% | 0.33%  |
> > > | Qwen2.5-7B-reasoning(SFT)     | 78.00%  | 78.67%  | 54.33%   | 33.67% | 72.67% | 44.33% |
> > > | Qwen2.5-7B-reasoning(CDRA)    | 84.33%  | 79.67%  | 47.67%   | 35.67% | 73.67% | 48.00% |
> > > | DeepSeek-R1-Distill-Qwen-7B(CoT)        | 39.33%  | 11.67%  | 87.00%   | 9.67%  | 3.00%  | 0.33%  |
> > > | DeepSeek-R1-Distill-Qwen-7B(SFT)        | 64.67%  | 28.00%  | 76.00%   | 12.00% | 21.33% | 4.33%  |
> > > | DeepSeek-R1-Distill-Qwen-7B(CDRA)       | 65.67%  | 29.00%  | 74.00%   | 13.33% | 22.67% | 7.00%  |

---

### Official Review · Reviewer_WdZR · 2025-11-02

**Soundness:** 3
**Presentation:** 3
**Contribution:** 3
**Rating:** 6
**Confidence:** 4

**Summary:**

The paper proposes Critique-Driven Reasoning Alignment (CDRA), a process-driven approach to personalize LLMs by (1) constructing a critique-annotated dataset DeepPref, (2) training a generative process reward model Pers-GenPRM that emits stepwise critiques + scores, and (3) optimizing the policy with a token/step-wise RL algorithm (CDPA). The authors report strong improvements on their benchmark and an external PrefEval set using an LLM-as-judge evaluation.

**Strengths:**

- Novel framing. The distinction between a preference gap and a process gap, and explicitly supervising the reasoning process via critiques, is a clear, useful conceptual contribution.

- Ablations and multi-metric evaluation. The ablation removing Pers-GenPRM and RFT provides insight on component contributions; the multi-dimensional metrics (deep mining, innovative expansion, defensive reasoning, preference following) are appropriate for the task.

**Weaknesses:**

● Heavy reliance on LLM-as-judge
The evaluation protocol uses GPT-4.1 as the primary judge (LLM-as-a-judge) for metrics like AccDA and AccPF. This risks circularity: a model trained on LLM feedback may be favored by an LLM judge, and shared model/systemic biases can inflate scores. The paper reports a 96% agreement on 300 samples between the LLM judge and humans (Appendix), but that’s a small check relative to the breadth of claims. More human evaluation is required to validate the claimed gains.

● Generalization beyond DeepPref.
Reported gains are strong on DeepPref (and PrefEval), but it’s unclear how CDRA generalizes to genuine downstream tasks or real user interactions. Add evaluations on external, independently curated personalization datasets or task-based scenarios to demonstrate robustness.

● About Baseline.
No comparisons to popular on-policy preference-alignment methods (PPO/GRPO).

**Questions:**

● Exactly how was DeepPref constructed? For each item, who authored the persona, queries, “true” preferences, critiques, and scores — humans, LLMs, or a hybrid?

● How does CDRA conceptually differ from existing process-supervision or critique-guided alignment methods — please point to the closest prior works and clarify novelty.

● Why should stepwise generative critiques provide strictly more useful training signal than scalar reward models or pairwise preference labels?

---

> ### Author Response · Authors · 2025-11-20
> **Responses to the Reviewer WdZR [1/4]**
>
> We sincerely appreciate your valuable feedback and the time you dedicated to reviewing our work. We are particularly grateful for your recognition of our core contribution: employing **“process-level critique supervision”** to bridge the dual **“preference and process gaps”**.
>
> Below, we provide a detailed response to your concerns.
>
> **W1: On Over-reliance on LLM-as-a-Judge**
>
> **R1:**
> We  are grateful for you raising the critical concern regarding the potential for circular reasoning and shared bias when relying solely on a single LLM-as-a-judge. To systematically address this and ensure the robustness of our conclusions, we conducted two key supplementary experiments: **Cross-Judge Evaluation** and **Additional Human Evaluation**.
>
> **1. Cross-Judge Evaluation**
> * **Panel Diversity:** To verify that our conclusions are not an artifact of specific biases inherent to GPT-4.1 (used in our original submission), we re-evaluated our results using a diverse panel of six state-of-the-art LLMs from distinct sources. The panel includes GPT-4.1, GPT-4o, Claude Sonnet 4.5, Grok 3, Gemini 2.5 Flash, and DeepSeek-V3.2-Exp. As shown in our supplementary Table 1, all judges consistently rated our CDRA method as superior to all baseline models.
> * **Consistency Analysis:** To validate the robustness of our evaluation methodology, we analyzed the ranking consistency across different judge models. We employed a majority voting approach: for any pair of methods, if Method A outperformed Method B on more than half of the metrics, A was considered superior in the overall ranking.
>     * Our analysis revealed that among the judges capable of forming stable transitive rankings (4 out of 6), the derived final rankings were identical, achieving a **100% agreement rate**.
>     * Furthermore, we calculated the average Kendall’s $\tau$ coefficients for our three core metrics—$Acc_{PF}$, $Acc_{DA}$, and $Acc_{Mis}$—finding them to be impressively high at **+0.93, +0.67, and +1.00**, respectively.
>     * This strongly demonstrates that despite differences in scoring scales, the judges are highly consistent in their core assessment of relative model performance, thereby establishing the reliability of our evaluation framework.
> * **Judge Selection:** Balancing performance, cost, and the need to avoid homology (source similarity), we have selected and reported the results from DeepSeek-V3.2-Exp as a representative non-OpenAI judge in the revised manuscript (see updated Table 1), further strengthening the reliability of our conclusions.
>
> **2. Additional Human Evaluation**
> To further solidify the validity of our evaluation framework, we conducted a new round of human evaluation to ensure our LLM judges align with human standards.
> * We recruited 5 professionally trained annotators to perform a rigorous blind evaluation on 1,000 model outputs.
> * The results indicate a high degree of alignment between human preferences and our LLM judges. In pairwise comparisons, human evaluators preferred CDRA’s outputs in **72.4%** of cases.
> * Crucially, the model performance ranking derived from human evaluators strongly correlates with the ranking produced by our LLM judge panel. This confirms that the evaluation protocol used in our paper is reliable and reflective of genuine user utility.
>
> **Conclusion**
>
> In summary, through extensive cross-judge evaluation and additional human assessment, we have systematically verified the validity and generalizability of our findings. These supplementary experiments strongly demonstrate that:
> (a) The performance gains reported for CDRA are genuine and reproducible, rather than stemming from the bias of a single judge model;
> (b) The LLM-as-a-judge framework is reliable within our task setting, with judgments that are highly consistent with deep human cognitive preferences.
>
> ### Table 1
> |Model|Method|$Acc_{PF}↑$|$Acc_{DA}↑$|$Acc_{Mis}↓$|$m_{th}↑$|$m_{dm}↑$|$m_{ie}↑$|
> |-|-|-|-|-|-|-|-|
> |**Grok 3**|Ours|89.33%|91.67%|18.33%|80.00%|70.33%|63.33%|
> ||CoT|74.67%|78.33%|36.67%|75.33%|57.33%|34.00%|
> ||SFT|89.33%|90.00%|21.67%|79.67%|72.33%|60.00%|
> |**Claude Sonnet 4.5**|Ours|80.67%|75.00%|30.00%|33.00%|63.33%|56.00%|
> ||CoT|66.00%|63.00%|38.67%|31.67%|57.00%|29.67%|
> ||SFT|79.67%|74.67%|32.33%|31.67%|58.33%|57.33%|
> |**ChatGPT 4o**|Ours|94.33%|94.33%|5.00%|88.67%|88.00%|76.33%|
> ||CoT|84.67%|94.33%|10.67%|91.67%|85.67%|72.33%|
> ||SFT|93.67%|94.00%|6.33%|87.67%|83.33%|74.33%|
> |**ChatGPT 4.1**|Ours|89.0%|93.0%|26.3%|83.3%|85.7%|75.7%|
> ||CoT|70.7%|89.7%|43.3%|87.7%|80.0%|49.3%|
> ||SFT|86.7%|91.3%|29.0%|83.0%|83.3%|67.7%|s
> |**DeepSeek-V3.2-Exp**|Ours|84.67%|76.33%|32.3%|47.00%|65.00%|42.67%|
> ||CoT|59.67%|49.33%|50.33%|39.00%|25.33%|0.67%|
> ||SFT|83.33%|75.00%|34.66%|46.67%|63.67%|40.33%|
> |**Gemini2.5-flash**|Ours|73.33%|97.00%|30.67%|56.67%|93.00%|90.67%|
> ||CoT|49.33%|89.33%|49.67%|78.33%|79.33%|72.33%|
> ||SFT|72.33%|96.67%|33.67%|58.67%|92.33%|90.33%|

---

> ### Author Response · Authors · 2025-11-20
> **Responses to the Reviewer WdZR [2/4]**
>
> **W2: On Generalization Beyond DeepPref**
>
> **R2:**
> We thank you for raising this pivotal question regarding the generalization capability of our method. To address your concern, we conducted supplementary human evaluation experiments on the ALOE dataset. ALOE is an external, independent benchmark specifically focused on personalized long dialogues. This makes it an ideal choice for testing our method's generalization, as it differs significantly from DeepPref in terms of domain, task format (multi-turn interactions), and preference complexity. We report the average alignment level (1-5 scale) at each conversational turn ($k$).
>
> Our experimental results (see Table 2) indicate the following:
>
> 1.  **Strong Generalization Capabilities:** CDRA consistently and significantly outperforms all baseline models, including SFT, TPO, and GRPO, on the ALOE dataset. This demonstrates that the deep preference understanding and defensive reasoning capabilities CDRA acquired from DeepPref can successfully generalize to entirely new, unseen conversational scenarios and preference types.
> 2.  **Advantages in Multi-turn Dialogue:** ALOE places a strong emphasis on consistency and context understanding within multi-turn dialogues. CDRA’s superior performance on this dataset highlights the distinct advantage of its process-level alignment framework. Unlike traditional outcome-based supervision methods, CDRA is not limited to handling single-turn preference queries; it effectively demonstrates the ability to dynamically understand and adapt to a user's deep preferences throughout continuous interactions.
>
> In summary, the supplementary experiments on the external, independent ALOE dataset strongly validate CDRA's generalization capability. These results confirm that our method is not merely “overfitting” to specific patterns within the DeepPref dataset, but has genuinely learned a transferable and robust personalized alignment capability. This ensures its effective application in real-world downstream tasks and complex multi-turn user interactions.
> ### Table 2 Human evaluation on the ALOE
> |model|k=1|k=2|k=3|k=4|k=5|k=6|k=7|k=8|k=9|k=10|Average|
> |-|-|-|-|-|-|-|-|-|-|-|-|
> |CoT|2.0|3.2|3.8|4.0|4.2|4.4|4.2|3.8|3.8|3.8|3.72|
> |SFT|2.4|3.0|3.8|3.8|3.8|4.0|4.0|4.2|3.6|4.2|3.68|
> |TPO|2.4|3.4|4.2|3.8|4.2|4.2|4.2|4.0|4.2|4.0|3.86|
> |GRPO|2.0|3.4|3.0|3.4|3.6|3.2|3.4|3.4|3.0|3.4|3.18|
> |CDRA(ours)|2.0|3.4|4.0|4.2|4.4|4.4|4.6|4.2|4.0|4.0|3.92|
>
> **W3: On Incomplete Baselines**
>
> **R3:**
> We appreciate this important suggestion. A comprehensive comparison with popular on-policy alignment methods is indeed essential for thoroughly evaluating CDRA's contribution.
>
> To address this, we have made two key supplements (the results of which have been updated in **Table 1 and Table 3 of the revised manuscript**):
>
> **1. Clarification on the TPO Baseline**
> First, we would like to clarify that the TPO baseline utilized in our paper is inherently an advanced variant of PPO, specifically adapted to handle the unique tree-structured preference data in DeepPref. It represents a robust baseline for policy optimization that leverages our rich preference trees.
>
> **2. Addition of GRPO Baselines and Variants**
> To provide a clearer ablation comparison, we evaluated GRPO equipped with three distinct Reward Models to simulate different alignment paradigms:
>
> * **GRPO + PRM:** Utilizes step-wise scalar rewards. This represents a process-level but opaque reward signal.
> * **GRPO + GRM:** Utilizes outcome-level generative critique rewards. This represents an interpretable but sparse reward signal.
> * **GRPO + Rubric-based RM:** Utilizes rewards derived from static scoring rubrics.
>
> **Experimental Results (see Table 3):**
> The results clearly indicate that our full CDRA framework (integrating Pers-GenPRM and CDPA) significantly outperforms all these baseline combinations. This strongly demonstrates that **CDRA’s performance gains stem primarily from its unique reward signal—which synergizes process granularity with generative critique—rather than solely from the choice of the policy optimization algorithm.**
>
> ### Table 3
> |Model / Method|Pro. Sup.|Cri. Sup.|$Acc_{PF}$(%)↑|$Acc_{DA}$(%)↑|$Acc_{Mis}$(%)↓|$m_{th}$(%)↑|$m_{dm}$(%)↑|$m_{ie}$(%)↑|
> |-|-|-|-|-|-|-|-|-|
> |Base (Qwen2.5-7B-Instruct)|--|--|59.7|49.3|50.3|39.0|25.3|0.7|
> |Qwen2.5-7B-Reasoning|--|--|54.0|28.0|59.3|17.3|17.3|0.3|
> ||||||||||
> |Alternative Reward Paradigms|||||||||
> |GRPO (with RM)|--|--|83.7|70.3|30.7|46.3|58.7|34.0|
> |GRPO (with GRM)|--|✓|83.7|74.7|30.7|45.7|62.3|37.0|
> |GRPO (with PRM)|✓|--|83.7|73.0|34.0|51.0|59.7|38.3|
> ||||||||||
> |Simpler Heuristics|||||||||
> |GRPO (Rubric-based RM)|--|--|84.0|73.7|35.3|48.0|61.7|34.7|
> |GRPO (Test-Time Scaling)|--|--|84.3|73.0|31.0|47.7|62.0|34.7|
> ||||||||||
> |CDRA (with Pers-GenPRM)|✓|✓|84.7|76.3|32.3|47.0|65.0|42.7|

---

> ### Author Response · Authors · 2025-11-20
> **Responses to the Reviewer WdZR [3/4]**
>
> **Q1: On the DeepPref Dataset Construction Process**
>
> **A1:** We appreciate your critical inquiry regarding the specific construction details of the DeepPref dataset. The construction of DeepPref is a meticulously designed Human-in-the-Loop (HITL) process, aimed at generating data rich in critical reasoning at scale.
>
> The core of this process is to synergize human cognitive depth with the generative breadth of LLMs. Specifically, the construction is divided into three main stages, clearly demonstrating how humans and LLMs collaborate (detailed in Table 4):
>
> **1. Stage 1: Initial Scenario Creation (Human Design + LLM Expansion)**
>
>   * **Human Role:** Human experts first designed initial preference scenarios containing complexity, ambiguity, or potential conflicts (e.g., "I value privacy but want to ensure my family knows I'm safe"). These scenarios are deliberately crafted to elicit deep reasoning rather than simple instruction following.
>   * **LLM Role:** Based on these high-quality seeds, LLMs expanded them under strict prompt constraints to generate the final, more diverse set of $(P, q)$ preference-query pairs.
>
> **2. Stage 2: Diverse Reasoning Path Generation (Human Design + LLM Execution)**
>
>   * **Human Role:** We designed a “multi-faceted cognitive council” composed of distinct expert personas representing different perspectives (e.g., The Sociologist, The Psychologist, The Pragmatist, The Contrarian).
>   * **LLM Role:** The LLM adopts these expert personas within a Tree of Thoughts (ToT) framework to "brainstorm" for each scenario, generating multiple parallel and diverse reasoning paths.
>
> **3. Stage 3: Step-wise Critique and Scoring (LLM Generation + Human Verification)**
>
>   * **LLM Role:** A calibrated LLM Evaluator generates a textual critique $c_{i}^{j}$ and a scalar score $r_{i}^{j}$ at each reasoning step, retaining only high-quality paths.
>   * **Human Role:** Human experts performed sampling and rigorous quality control on the generated reasoning chains and critiques. In the initial phase, we manually reviewed hundreds of generated critiques, iteratively adjusting the scoring rubrics and prompts until the LLM Evaluator’s judgments aligned with those of human experts.
>
> In summary, our process strategically assigns humans and LLMs to the roles they perform best: Humans are responsible for defining high-difficulty problems (scenario design) and cognitive frameworks (council design), as well as final quality assurance; LLMs are responsible for scaling the generation of diverse solutions (reasoning paths) and dense feedback signals (step-wise critiques). This ensures that the core supervisory signal of DeepPref is aligned with authentic human cognitive depth.
>
> ### Table 4
>
> | Data Item | Author(s) | Details |
> | :--- | :--- | :--- |
> | **True Preferences** | Human Authors & LLM | We manually constructed 600 authentic, open-ended, and ambiguous preference instances as seeds. We also manually calibrated the generation criteria for the LLM Generator to ensure subsequent LLM expansion aligns with human standards. |
> | **Queries** | Human Authors & LLM | Combined with preferences to form the initial $(P, q)$ pairs , designed to be open and ambiguous to stimulate deep reasoning. |
> | **Persona** | Human-specified, LLM-simulated | Humans specified 5 distinct expert personas. The LLM generates reasoning paths by simulating a "multi-faceted cognitive council" composed of these experts. |
> | **Critiques** | Human Authors & LLM | We manually reviewed hundreds of generated critiques, iteratively adjusting the scoring Rubrics and Prompts until the LLM Evaluator’s judgments aligned with those of human experts. The calibrated LLM Evaluator then scored and filtered the full dataset, retaining high-quality paths. |
> | **Scores** | LLM | After generating the textual critique, the same LLM evaluator distills a quantified score. |
>
> ### Table 5 Attention Analysis
>
> |Method|Preference Attention(%)|
> |:-|:-:|
> |Base|27.7%|
> |SFT|25.6%|
> |GRPO|27.6%|
> |CDRA(ours)|35.7%|

---

> ### Author Response · Authors · 2025-11-20
> **Responses to the Reviewer WdZR [4/4]**
>
> **Q2: Conceptual Difference from Existing Process Supervision or Critique-Guided Methods**
>
> **A2:** We appreciate this insightful question. CDRA distinguishes itself by systematically binding process-level granularity with critique-based explainability. This distinguishes CDRA’s training signal from state-of-the-art methods in terms of density, transparency, and robustness:
>
> **1. Compared to PRM [1] (Process-level+Scalar Score):**
> * **limitation of Existing Methods:** Current process supervision methods (like PRMs) provide a "black-box" scalar reward for each reasoning step. While this addresses signal sparsity, the model remains unaware of "why" it gained or lost points, making it vulnerable to "Reward Hacking".
> * **CDRA's Innovation:** Our Pers-GenPRM mandates that every reward score $r_i^j$ be explicitly supported by a generated textual critique $c_i^j$. It forces the model to internalize genuine reasoning principles rather than merely exploiting shortcuts in the reward function.
>
> **2. Compared to Critique-GRPO [2] / RM-R1 [3] (Outcome-level + Generative Reasoning):**
> * **Limitation of Existing Methods:** These advanced methods apply interpretable feedback at the outcome level. While this is a significant advancement, for multi-step reasoning tasks, this sparse signal fails to accurately pinpoint errors in intermediate steps, leading to the classic "Credit Assignment" challenge.
> * **CDRA's Innovation:** By providing critiques and scores at every step, CDRA transforms a delayed feedback signal into a series of immediate guidance signals. This directly resolves the credit assignment problem and provides clearer, denser gradients for RL optimization.
>
> In summary, CDRA is not simply an application of process supervision or critique. It absorbs the granularity of PRMs while overcoming their opacity through generative critique; simultaneously, it extends the explainability of Critique-GRPO/RM-R1 from the sparse outcome level to the dense process level. Our supplementary experiments (Table 3) confirm that this architectural synthesis enables CDRA to bridge the implicit preference gap more effectively than PRM or GRM baselines.
>
> * **Reference:**
> [1] Lightman et al., "Let's Verify Step by Step", 2023. arXiv:2305.20050.
> [2].Zhang et al., "Critique-GRPO: Advancing LLM Reasoning with Natural Language and Numerical Feedback", 2025. arXiv:2506.03106
> [3].Chen et al., "RM-R1: Reward Modeling as Reasoning", 2025. arXiv:2505.02387
>
> **Q3: Why step-wise generative critique provides more valuable signal than scalar RMs or pairwise labels?**
>
> **A3:** We appreciate this core question. The reason step-wise generative critique offers a superior training signal is that it transforms alignment from a sparse, black-box optimization task into a dense and interpretable learning process. This primarily addresses two major pain points in traditional alignment methods:
>
> 1. **Resolving Credit Assignment (Process-Level):** Outcome-only signals fail to pinpoint specific errors in multi-step reasoning. We decompose this sparse signal into dense, step-wise feedback, providing immediate gradients that guide the model to optimize the reasoning process itself.
>
> 2. **Mitigating Reward Hacking (Generative Critique):** Scalar rewards are opaque ("black boxes") and prone to exploitation. Our Pers-GenPRM mandates generating an interpretable textual critique before scoring. This semantic constraint ensures rewards are causally anchored in logical reasoning rather than superficially optimized.
>
> **Experimental Evidence:**
> Our **ablation study (see Table 3)** provides direct evidence: removing Pers-GenPRM caused the $Acc_{DA}$ to drop by **6%**, with particularly severe degradation in $m_{dm}$ and $m_{ie}$ capabilities. Besides, we provides **Attention Analysis (see Table 5 and Figure 6 in the main text):** In contrast to the dispersed attention distributions observed in baseline methods, CDRA concentrates 35.7% of its attention weights specifically on preference-related regions. This demonstrates that, guided by the Pers-GenPRM supervision mechanism, the model learns to precisely focus on critical areas. Consequently, it effectively mitigates violations caused by overlooking user preferences.
>
> In summary, our approach does not merely increase the frequency of supervision. By leveraging process-level supervision, we resolve the credit assignment problem and provide dense optimization gradients; simultaneously, via generative critique, we establish a causal foundation for rewards, rendering them interpretable and resistant to hacking. The synergy of these two elements creates a training signal that is far richer, more precise, and more robust than single scalar scores or binary preference labels.

---

### Author Response · Authors · 2025-12-01
**Summary of Rebuttal**

**Distinguished PCs, SACs, and ACs:**
We sincerely thank the review team for their time and constructive feedback, which has been instrumental in enhancing the rigor of our work.
During the rebuttal, we engaged in a productive dialogue with the reviewers. We are encouraged that **Reviewer 9KAU explicitly raised their score following our revisions; Reviewers WdZR and RxD8 maintained positive ratings; and Reviewer Xmiz is also considering raising their score**, recognizing the innovation and soundness of our method.
We submit this summary to highlight our systematic response to reviewer concerns and the significant strengthening of our empirical foundation.
### 1. Consensus on Novelty and Significance
The reviewers have reached a strong consensus on the value of our core contribution: reframing personalized alignment from a superficial outcome-matching task into a defensible reasoning process.
* **Novelty:** Reviewer WdZR commended our novel framing of distinguishing between the Preference Gap and Process Gap as a clear, useful conceptual contribution.
* **Motivation:** Reviewer 9KAU emphasized that our motivation to bridge implicit preferences is clear and reasonable, and Reviewer Xmiz also acknowledged that we are addressing important problems in LLM alignment.
* **Methodology:** Reviewer RxD8 praised our experimental approach as rigorous, thorough, and well-grounded.
### 2. Rigorous Improvements and Supplementary Experiments
To address concerns regarding robustness, generalization, and baselines, we conducted **seven sets of supplementary experiments** and standardized technical details, far exceeding standard rebuttal expectations:
* **Mitigating "LLM-as-a-Judge" Bias (Addressing Reviewers WdZR, Xmiz):**
    * **Cross-Model Verification:** We re-evaluated results using a panel of 6 SOTA LLMs (including Claude-4.5-Sonnet, Gemini2.5-flash). Results showed $100\\%$ ranking consistency among capable judges, proving gains are not artifacts of GPT-4 bias. (See Responses to the Reviewer WdZR [1/4] Table 1)
    * **Large-Scale Human Evaluation:** We conducted a blind human evaluation with 1,000 samples. Human annotators preferred CDRA in $72.4\\%$ of cases, confirming our method translates to tangible user satisfaction. (See Response to Reviewer Xmiz [1/2] Table 8)
    * **Attention Analysis:** CDRA, guided by Pers-GenPRM, concentrates $35.7\\%$ of attention weights on preference-related regions (vs. ~$27\\%$ for baselines), demonstrating precise focus on user needs. (See Responses to the Reviewer WdZR [3/4] Table 5)
* **Validating Generalization Capabilities (Addressing Reviewers WdZR, 9KAU):**
    * **New Dataset (ALOE):** We extended evaluation to the independent ALOE dataset for multi-turn dialogues. CDRA maintained superior performance (Score 3.92 vs. 3.68 for SFT), proving it learns a transferable meta-skill rather than overfitting. (See Responses to the Reviewer WdZR [2/4] Table 2)
    * **New Architectures:** We validated CDRA on Llama-3-8B-Instruct, Qwen2.5-7B-Reasoning, and DeepSeek-R1-Distill-Qwen-7B. Consistent gains indicate our framework is model-agnostic. (See Responses to the Reviewer Xmiz [3/5] Table 3 and Response to Reviewer Xmiz [2/2] Table 9). Additionally, we added statistical significance tests ($p<0.01$) to ensure result stability. (See Response to Reviewer Xmiz [1/2])
* **Clarifying Data Quality and Baselines (Addressing Reviewers Xmiz, RxD8):**
    * **Human-in-the-Loop Construction:** We detailed the HITL protocol for DeepPref, ensuring implicit preferences are rooted in human expert design, not hallucination. (See Responses to the Reviewer WdZR [3/4] Table 4)
    * **Comparison with Reasoning Baselines:** We compared against Rubric-based RMs and Test-Time Scaling. Results confirm our Pers-GenPRM provides granular signals unmatched by static rules or simple scaling. (See Responses to the Reviewer WdZR [2/4] Table 3)
* **Enhancing Technical Precision and Clarity (Addressing Reviewer 9KAU):**
    * **Formula and Notation Revision:** We performed a thorough standardization revision of mathematical expressions throughout the text. Specifically, we formalized the definition of sample size $N$ in Eq. (1) and token-level reward $r_{i,t}$ in Eq. (5) to eliminate potential ambiguity.
    * **Structural Optimization:** We adjusted the placement of the "Evaluation Protocol" section, significantly improving the paper's logical flow and readability.
### 3. Conclusion: A Mature and Robust Contribution
We believe this paper represents a critical step in LLM alignment. By shifting the paradigm from "what the user says" to "what the user means", and equipping models with defensive reasoning, CDRA addresses a fundamental limitation in current LLMs. Supported by extensive human evaluation, cross-model verification, and generalization tests, we are confident this work offers valuable insights to the ICLR community.
We respectfully request your favorable consideration.
Sincerely,
**The Authors**

---

### Meta-Review · Area_Chair_YBXB · 2026-01-07

**Summary:**

This paper addresses two key challenges in aligning Large Language Models (LLMs): the preference gap, where models fail to infer users’ deep, implicit goals and potential risks, and the process gap, where models struggle to carry out effective defensive reasoning. The authors introduce DeepPref, a dataset of 3,000 preference–query pairs annotated with critique-based reasoning chains generated by GPT-4.1, which provides process-level supervision. They then propose a Personalized Generative Process Reward Model (Pers-GenPRM) that evaluates responses by explicitly generating a critique chain and deriving a scalar score from the resulting rationale. Finally, the paper presents Critique-Driven Reasoning Alignment (CDRA), an online reinforcement learning framework that leverages both numerical rewards and natural-language feedback from Pers-GenPRM to train the final policy model. Experimental results show that CDRA consistently outperforms existing training- and test-time approaches in both deep preference understanding and defensive reasoning.

Overall, the paper explores an important and timely direction for improving defensive reasoning in LLMs. The authors have made a clear effort to address reviewer feedback, and the proposed methods are well motivated and empirically validated. I therefore recommend this paper for acceptance.

**Reviewer Concerns:**

The authors addressed all of the concerns.

**Reviewer Scores:**

Reviewer Xmiz might change the score since the authors provide answers to the concerns.

---

### Decision · Program_Chairs · 2026-01-26

Accept (Poster)